# Prognostic Value of Systemic Immune-Inflammation Index for Major Adverse Cardiac Events and Mortality in Severe Aortic Stenosis Patients after TAVI

**DOI:** 10.3390/medicina57060588

**Published:** 2021-06-08

**Authors:** Aydin Rodi Tosu, Muhsin Kalyoncuoglu, Halil İbrahim Biter, Sinem Cakal, Murat Selcuk, Tufan Çinar, Erdal Belen, Mehmet Mustafa Can

**Affiliations:** 1Department of Cardiology, Haseki Training and Research Hospital, University of Health Sciences, 34130 Istanbul, Turkey; aydinroditosu@gmail.com (A.R.T.); mkalyoncuoglu80@gmail.com (M.K.); abrahambiter@gmail.com (H.İ.B.); Sinemdnz@gmail.com (S.C.); belenerdal@gmail.com (E.B.); mehmetmustafacan@yahoo.com (M.M.C.); 2Department of Cardiology, Sultan II. Abdülhamid Han Training and Research Hospital, University of Health Sciences, 34668 Istanbul, Turkey; murat-slck@hotmail.com

**Keywords:** SII, inflammation, major adverse cardiac events, mortality, TAVI

## Abstract

*Background and objectives:* In this study, we aimed to evaluate whether the systemic immune-inflammation index (SII) has a prognostic value for major adverse cardiac events (MACEs), including stroke, re-hospitalization, and short-term all-cause mortality at 6 months, in aortic stenosis (AS) patients who underwent transcatheter aortic valve implantation (TAVI). *Materials and Methods:* A total of 120 patients who underwent TAVI due to severe AS were retrospectively included in our study. The main outcome of the study was MACEs and short-term all-cause mortality at 6 months. *Results:* The SII was found to be higher in TAVI patients who developed MACEs than in those who did not develop them. Multivariate Cox regression analysis revealed that the SII (HR: 1.002, 95%CI: 1.001–1.003, *p* < 0.01) was an independent predictor of MACEs in AS patients after TAVI. The optimal value of the SII for MACEs in AS patients following TAVI was >1.056 with 94% sensitivity and 96% specificity (AUC (the area under the curve): 0.960, *p* < 0.01). We noted that the AUC value of SII in predicting MACEs was significantly higher than the AUC value of the C-reactive protein (AUC: 0.960 vs. AUC: 0.714, respectively). *Conclusions:* This is the first study to show that high pre-procedural SII may have a predictive value for MACEs and short-term mortality in AS patients undergoing TAVI.

## 1. Introduction

Severe aortic stenosis (AS) can be caused by rheumatic heart disease or more commonly by the calcification of a congenitally bicuspid or trileaflet aortic valve [1]. The prevalence of AS increases with age, averaging 0.2% at 50–59 years and increasing to 9.8% at 80–89 years [1]. Moreover, in the past few years, severe AS had been identified as an inflammatory-based disease sharing a common pathophysiology with atherosclerosis, such as deposition of lipoproteins, calcification, and chronic inflammation [2].

Currently, transcatheter aortic valve implantation (TAVI) is an alternative treatment option to surgical aortic valve implantation in patients with inoperable and high-risk severe symptomatic AS [3]. Prior studies revealed that several biomarkers, such as galectin-3, growth differentiation factor-15, microRNAs, and brain natriuretic peptides, were known to predict outcomes after TAVI [4,5]. Moreover, in a meta-analysis of 14 studies including 3449 TAVI patients, high-baseline inflammatory status, assessed by C-reactive protein (CRP) levels, may predict increased mid-term mortality following TAVI [5]. 

Recently, the systemic immune-inflammation index (SII) was developed based on the *n* × *p*/L (*n*, *p* and L represent neutrophil counts, platelet counts and lymphocyte counts, respectively) ratio to consider patients’ inflammatory and immune status simultaneously [6]. Remarkably, high SII has been reported to be associated with poor outcomes in cancer patients [7]. Moreover, this index has been shown to have a predictive value for mortality in patients with cardiovascular disease, including coronary artery disease and acute coronary syndrome [8,9]. However, the predictive value of the SII for MACEs in AS after TAVI is unknown. Since the SII is a non-invasive, widely available, and easily obtained test as part of a complete blood count (CBC), we aimed to study whether this index might have a prognostic value for MACEs, including stroke, re-hospitalization, and short-term all-cause mortality at 6-months, in AS patients undergoing TAVI.

## 2. Materıal and Methods

### 2.1. Study Population

This retrospective study was carried out in a total of 120 symptomatic severe AS patients who underwent TAVI at a tertiary heart center between June 2014 and June 2020. The exclusion criteria were as follows: (I) evidence of acute or chronic infection, (II) systemic inflammatory or autoimmune disease, (III) a history of any liver disease (with liver function parameters more than three times the upper normal limit), (IV) any clinically significant endocrine, hematologic, respiratory, or metabolic diseases and (V) malignancy. In all of the patients, clinic and laboratory data were retrieved from the hospital’s electronic database. Informed consent for each case before TAVI was obtained. The selection of patients with severe symptomatic AS was based on the expected perioperative or short-term mortality that was estimated from the risk model of the Society of Thoracic Surgeons Score (STS) algorithm. All patients were evaluated by our multidisciplinary heart team before the TAVI procedure because it is well-known that care managers play an important role in this context [10]. Patients with severe AS were considered as a candidate for TAVI after being determined as a high or very high cardiac surgical risk. Procedural complications were defined using valve academic research consortium 2 (VARC-2) criteria [11].

### 2.2. SII

In all patients, blood samples were collected from the antecubital vein before TAVI. Complete blood cell counts, including platelet, neutrophil, and lymphocyte, were determined with an auto-analyzer. The SII was calculated as total peripheral platelets count (*p*) × neutrophil-to-lymphocyte ratio (*n*/L) (SII = *p* × *n*/L ratio) [6].

### 2.3. Definitions

MACEs were defined as stroke or transit ischemic attack, re-hospitalization, and all-cause short-term mortality at 6 moths. An electronic database of the hospital and a state-wide death registry database were used for the evaluation and confirmation of medium-term deaths. Stroke was considered as a TIA if it involved a neurological deficit of <24 h or a stroke if it was longer. Re-hospitalization was defined as readmission within 30 days.

### 2.4. Statistical Analysis

All statistical analyses were performed using the Statistical Package for the Social Sciences version 24.0 software program (IBM Corp., Armonk, NY, USA). All categorical variables were given as frequencies and (%). The chi-square (χ^2^) test was used to compare the categorical variables. Continuous variables were given as mean ± standard deviation (if normal distribution) and medians (interquartile ranges (IQR)) (if not normal distribution). The Kolmogorov–Smirnov test was used to assess whether the variables were normally distributed. Student’s t-test or the Mann–Whitney *U* test were used to compare continuous variables regarding whether they were normally distributed or not. In order to identify the independent risk factors for MACEs, univariable and multivariable Cox regression analyses were performed with the enter method. Only the variables with a *p* value of less than 0.05 in the univariable analysis were incorporated in the multivariate Cox regression analysis. To prevent multi-collinearity between SII and CRP, two models were created to determine the independent predictors of MACEs. The results of the Cox regression analysis are reported with hazard ratios (HRs) and 95% confidence intervals (CIs). The receiver operating characteristics (ROC) curve analysis was used to evaluate the sensitivity and specificity of the SII and CRP for predicting MACEs. The area under the curve (AUC) was used as a measure of the predictive accuracy of SII and CRP. Survival evaluation for the low- and high-SII groups was undertaken using Kaplan–Meier and the log-rank test. The results were evaluated within a 95% CI and at a significance level of *p* < 0.05.

## 3. Results

The mean age of the study population was 77.5 ± 4.6 years, and 68 (56.7%) patients were male. We divided the study population into two groups: patients with and without MACEs. The demographic characteristics and medications of all patients are summarized in Table 1.

The prevalence of comorbidities, including hypertension, diabetes, smoking, hyperlipidemia, peripheral disease, chronic obstructive lung disease, and atrial fibrillation, was not different between the groups. Additionally, the histories of coronary artery disease, chronic renal failure, and cerebrovascular accident were similar between groups. Previous medications were also not different between the groups.

In terms of laboratory variables, fasting blood glucose, creatinine, lipid profiles, hemoglobin, and platelet counts were similar in both groups (Table 2).

We observed that patients who developed MACEs had higher CRP and white blood cell counts. Remarkably, patients with MACEs also had higher SIIs than those without MACEs (1283 (IQ = 1153–1399) vs. 522 (IQ = 344–845), *p* < 0.01, respectively).

The preprocedural and procedural features of all cases based on the median SII are summarized in Table 3.

Patients with high SIIs had lower aortic valve areas and mean aortic valve gradients (AVG) compared to those with low SII. Other procedural features, such as left ventricular ejection fraction, STS score, predilation, postdilatation, implantation depth, and type of valve (balloon-expandable or self-expandable), were not different between the groups.

Procedural and postprocedural complications of all cases based on the SII are displayed in Table 4.

The short-term all-cause mortality at 6 months was 6.7%. In all cases, the major vascular complications and major bleeding rates were 15% and 7.5%, respectively. Patients with high SIIs had higher incidences of major vascular complications, major bleeding, permanent pacemaker implantation, re-hospitalization, post-procedural ischemic stroke and transient ischemic attack than those with low SII.

In order to determine the predictors of MACEs, we performed both univariable and multivariable Cox regression analyses. The SII, CRP, STS, and AVG were predictors of MACEs in univariable regression analysis. We generated two multivariable Cox regression analysis models in order to prevent multi-collinearity between SII and CRP. In the multivariable model-1, only AVG and STS were independent predictors for MACEs. According to the multivariable model-2, only SII (HR: 95%CI; 1.002 (1.001–1.003), *p* < 0.01) and STS were independent predictors for MACEs. Interestingly, CRP was not independently related with MACEs according to our multivariate regression analysis (Table 5).

In an ROC curve analysis, the cut-off value of the SII level in predicting MACEs was determined as greater than 1.056, with 94% sensitivity and 96% specificity (AUC: 0.960; 95%CI: 0.907–0.987, *p* < 0.01) (Figure 1). We noted that the AUC of SII in predicting MACEs was significantly higher than the AUC of CRP (AUC: 0.960 vs. AUC: 0.714, respectively) (Figure 1).

The Kaplan–Meier analysis revealed that patients with high SII had higher mortality rates than those with low SII in the short-term follow-up (Figure 2).

## 4. Discussion

This is the first study to investigate whether there is a relationship between SII, MACEs and short-term mortality in AS patients undergoing TAVI. We have shown that increased SII is directly related to the incidence MACEs and short-term mortality in such patients.

Endothelial dysfunction and chronic inflammation, which play critical roles in the pathogenesis of atherosclerosis, are also observed in the pathogenesis of severe AS patients [12,13]. Currently, cardiac valve surgery is the recommended treatment strategy for AS patients with low or moderate surgical risk [3]. On the other hand, the TAVI is most often performed in high-risk patients. Traditional risk scores for TAVI (e.g., STS, European System for Cardiac Operative Risk Evaluation (EuroSCORE)) do not include some important parameters, such as nutritional status, frailty, and inflammation; all of them are considered as important risk factors for increased mortality in AS patients undergoing TAVI [14,15]. Hence, it is clear that additional parameters are needed in the prognostic evaluation of such patients.

Prior studies showed that chronic inflammation might be related to increased mortality in AS patients following TAVI [5]. Condado et al. showed that neutrophil–lymphocyte ratio and platelet–lymphocyte ratio could be used to risk-stratify AS patients after TAVI [16]. Moreover, Iglesias-Álvarez et al. found that hs-CRP at baseline was an independent predictor of post-TAVI mortality [17]. Additionally, Okuno et al. reported that patients with a lower prognostic nutritional index (PNI) and a higher controlling nutritional status score had a significantly higher composite outcome of 1-year mortality as well as re-hospitalization due to heart failure [18]. In our research, we observed that CRP levels were not independently linked with MACEs in AS cases following TAVI. Remarkably, our findings also clearly revealed that the AUC of SII was significantly higher compared to the AUC of CRP levels.

The SII, which can be used to determine the inflammatory status of the patient, is calculated based on the platelet, neutrophil, and lymphocyte counts [6]. As this index is simple and easily calculable, it is considered that it can be used as an effective parameter for prognostic information in patients with various solid tumors [1,19]. Furthermore, in a recent study, Seo et al. reported the predictive value of SII for mortality in patients with chronic heart failure [20]. Besides that, the prognostic importance of this index has been shown in patients with various cardiovascular diseases [8,9]. However, in the current literature, there is no prior clinical study demonstrating the prognostic value of SII for MACEs and short–term mortality in severe AS patients after TAVI. The present study might be the first to demonstrate that the SII may have a good predictive value for predicting MACEs in AS patients undergoing TAVI. Additionally, the SII was independently related to poor cardiovascular events and short-term mortality in such patients after TAVI. Our study findings provide evidence for the association between high SII, increased MACEs and short-term mortality among these patients after TAVI. These findings suggest that the inflammatory status as determined by SII is an effective parameter in predicting MACEs in such patients after TAVI.

We considered that our results are important for the management of severe AS before TAVI. As a simple and inexpensive index that can be easily calculated from CBC parameters, the SII can used to stratify severe AS patients that are more likely to benefit from TAVI. In addition, our findings provided evidence that closer follow-up should be arranged after TAVI in patients with high SII.

### Study Limitations

Our study had some limitations. First, the study had a retrospective and observational design, which could be accepted as the major limitation of the study. Second, this study had a small sample size, and the follow-up period was short. Therefore, the limited number of patients with a short-term follow-up period might prevent the generalizability of our study. Third, in the current investigation, a spot laboratory value was used to estimate short-term mortality. Fourth, despite using a multivariable analysis, there might be some unmeasured confounders, all of which can affect the study results. Finally, our findings warrant prospective and multicenter studies with larger sample sizes to elucidate the relationship between the SII, MACEs and short-tern mortality in AS patients following TAVI.

## 5. Conclusions

In the present investigation, we have shown that the SII can be used as an effective inflammatory parameter for TAVI outcomes, and specifically for MACEs and short-term mortality, in severe AS patients. As a cheap and easily calculable inflammatory parameter, the SII can be used to determine such events in AS patients following TAVI.

## Figures and Tables

**Figure 1 medicina-57-00588-f001:**
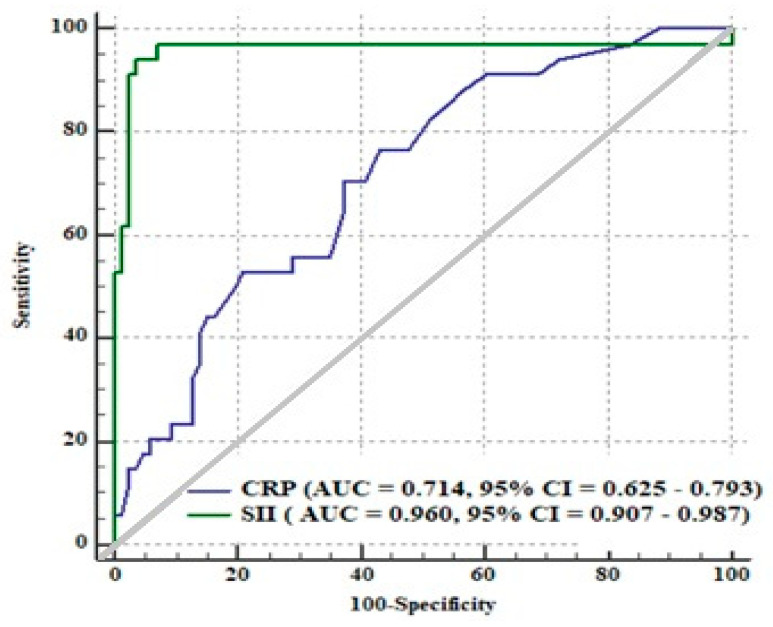
A receiving operating characteristics curve analysis comparison of the systemic immune-inflammation index (SII) and C-reactive protein for major adverse cardiac events (MACEs), including stroke, re-hospitalization, and short-term all-cause mortality at 6 months. CRP: C-reactive protein; AUC: the area under the curve.

**Figure 2 medicina-57-00588-f002:**
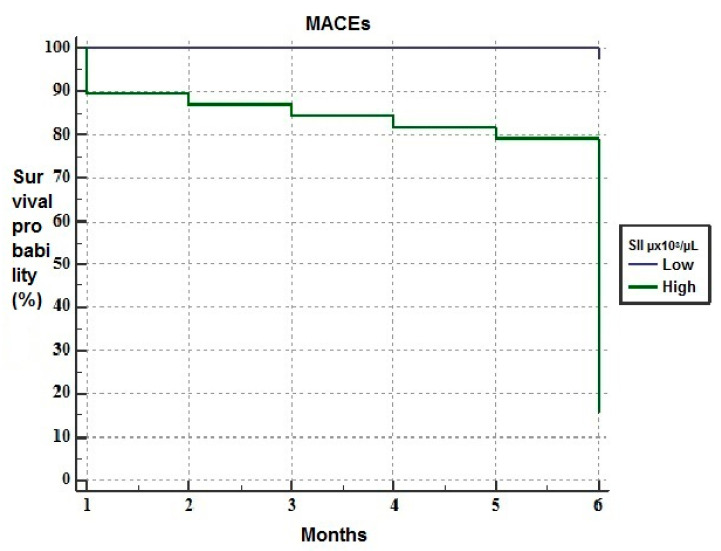
The Kaplan–Meier plots of survival curves of patients with low (blue line) and high systemic immune-inflammation index (SII) (green line).

**Table 1 medicina-57-00588-t001:** Demographic and clinical parameters of the study cohort.

	All Population (*n* = 120)	MACEs (−) (*n* = 86)	MACEs (+) (*n* = 34)	*p* Value
Male gender, *n* (%)	68 (56.7)	47 (54.7)	21 (61.8)	0.48
Age, years	77.5 ± 4.6	77.6 ± 4.9	77.2 ± 3.7	0.70
BMI, kg/m^2^	23.1 ± 2.0	23.0 ± 2.0	23.2 ± 2.1	0.63
Risk factors				
Hypertension, *n* (%)	68 (56.7)	53 (61.6)	15 (44.1)	0.08
Diabetes, *n* (%)	72 (60)	50 (58.1)	22 (64.7)	0.51
Smoking, *n* (%)	16 (13.3)	12 (14.0)	4 (11.8)	0.75
Hyperlipidemia, *n* (%)	53 (44.2)	39 (45.3)	14 (41.2)	0.68
CAD history, *n* (%)	47 (39.2)	35 (40.7)	12 (35.3)	0.59
PAD, *n* (%)	28 (23.3)	22 (25.6)	6 (17.6)	0.35
CRF history, *n* (%)	23 (19.2)	16 (18.6)	7 (20.6)	0.80
COPD, *n* (%)	53 (44.2)	38 (44.2)	15 (44.1)	0.99
CVA history, *n* (%)	6 (5)	3 (3.5)	3 (8.8)	0.23
NYHA Class III-IV, *n* (%)	43 (35.8)	28 (32.6)	15 (44.1)	0.23
Atrial fibrillation, *n* (%)	20 (16.7)	12 (14)	8 (23.5)	0.20
Presence of BBB, *n* (%)	19 (15.8)	11 (12.8)	8 (23.5)	0.15
Medications				
Acetylsalicylic acid, *n* (%)	33 (27.5)	26 (30.2)	7 (20.6)	0.29
OAC, *n* (%)	16 (13.3)	10 (11.6)	6 (17.6)	0.38
Beta-blockers, *n* (%)	44 (36.7)	34 (39.5)	10 (29.4)	0.30
RAS blockers, *n* (%)	39 (32.5)	30 (34.9)	9 (26.5)	0.38
Statin, *n* (%)	18 (15)	12 (14)	6 (17.6)	0.61
Diuretic, *n* (%)	34 (28.5)	25 (29.1)	9 (26.5)	0.78

Continuous variables are presented as mean ± SD or median (IQR), nominal variables presented as frequency (%). Abbreviations: MACEs, major adverse cardiac events; BMI, body mass index; CAD, coronary artery disease; PAD, peripheral arterial disease; CRF, chronic failure; COPD, chronic obstructive pulmonary disease; CVA, cerebrovascular accident; NYHA, New York Heart Association; BBB, bundle branch block; OAC, oral anticoagulant; RAS, renin–angiotensin system.

**Table 2 medicina-57-00588-t002:** Laboratory data of the study cohort.

	All Population (*n* = 120)	MACEs (−) (*n* = 86)	MACEs (+) (*n* = 34)	*p* Value
FBG, mg/dL	125.8 ± 39.8	125.7 ± 40.5	126 ± 38.7	0.98
Creatinine, mg/dL	1.1 (0.8–1.58)	1.0 (0.78–1.53)	1.1 (0.8–1.83)	0.47
TC, mg/dL	206.5 ± 39.9	207.7 ± 41.9	203.4 ± 34.6	0.60
LDL-C, mg/dL	138.9 ± 39.1	139.2 ± 41.0	138.1 ± 33.6	0.89
HDL-C, mg/dL	41.2 ± 4.0	41.4 ± 6.9	40.6 ± 6.4	0.52
Triglyceride, mg/dL	175 (146.3–195)	175 (148.8–195.3)	174.5 (143–195)	0.35
CRP, mg/dL	2.5 (0.9–4.9)	1.8 (0.8–4.5)	4.9 (2.3–7.2)	<0.01
Hemoglobin, mg/dL	15.1 ± 3.5	14.9 ± 1.1	15.0 ± 3.0	0.86
Platelet, µ × 10^3^/µL	263.7 ± 73.2	260.4 ± 70	271.9 ± 81.4	0.44
WBC, µ × 10^3^/µL	7.0 (5.5–8.2)	6.2 (5.2–7.4)	8.3 (7.5–9.1)	<0.01
SII, 10^3^/µL	737 (407–1145)	522 (344–845)	1283 (1153–1399)	<0.01

Continuous variables are presented as mean ± SD and median (IQR). Abbreviations: MACEs, major adverse cardiac events; FBG, fasting blood glucose; TC, total cholesterol; LDL, low-density lipoprotein cholesterol; HDL, high-density lipoprotein cholesterol; CRP, C-reactive protein; WBC, white blood cell count; SII, systemic immune-inflammation index.

**Table 3 medicina-57-00588-t003:** Preprocedural and procedural characteristic of all cases according to the SII.

	All Population (*n* = 120)	Low SII (*n* = 82)	High SII (*n* = 38)	*p* Value
Aortic valve area, cm^2^	0.81 ± 0.1	0.83 ± 0.09	0.78 ± 0.1	0.01
Mean AVG, mmHg	47.8 ± 4.1	46.9 ± 3.3	50.0 ± 5.2	<0.01
LV ejection fraction, %	55.5 ± 7.9	55.4 ± 7.9	55.9 ± 8.2	0.77
STS score, median	9.6 (8.1–11.5)	10.3 (8.1–11.5)	9.2 (8.1–12.3)	0.96
Predilation, *n* (%)	19 (15.8)	15 (18.3)	4 (10.5)	0.28
Postdilatation, *n* (%)	16 (13.3)	9 (11)	7 (18.4)	0.26
Implantation depth, mm	5.2 ± 0.8	5.3 ± 0.7	5.1 ± 0.8	0.35
Type of valve, *n* (%)				
Balloon-expandable	37 (30.8)	28 (34.1)	9 (23.7)	0.25
Self-expandable	83 (69.2)	54 (65.9)	29 (76.3)	0.25

Continuous variables are presented as mean ± SD or median (IQR), nominal variables are presented as frequency (%). Abbreviations: AVG, aortic valve gradient; STS, Society of Thoracic Surgeons Score; LV, left ventricle; SII, systemic immune-inflammation index.

**Table 4 medicina-57-00588-t004:** Procedural and postprocedural complications of all cases according to the SII.

	All Population (*n* = 120)	Low SII (*n* = 82)	High SII (*n* = 38)	*p* Value
Major vascular complications, *n* (%)	18 (15)	1 (1.2)	17 (44.7)	<0.01
Major bleeding, *n* (%)	9 (7.5)	1 (1.2)	8 (21.1)	<0.01
Permanent pacemaker, *n* (%)	7 (5.8)	1 (1.2)	6 (15.8)	<0.01
Re-hospitalization, *n* (%)	9 (7.5)	0 (0)	9 (23.7)	<0.01
Postprocedural IS or TIA, *n* (%)	8 (6.7)	1 (1.2)	7 (18.4)	<0.01
6-month mortality, *n* (%)	8 (6.7)	0 (0)	8 (23.5)	<0.01
MACEs, *n* (%)	34 (28.3)	2 (2.4)	32 (84.2)	<0.01

Nominal variables presented as frequency (%). Abbreviations: MACEs, major adverse cardiac events; IS, ischemic stroke; TIA, transit ischemic attack; SII, systemic immune-inflammation index.

**Table 5 medicina-57-00588-t005:** Univariable and multivariable Cox proportional hazards regression analysis models for determining the predictors of the MACEs at 6 months.

	Univariable		Multivariable Model-1		Multivariable Model-2	
	HR (95%CI)	*p* Value	HR (95%CI)	*p* Value	HR (95%CI)	*p* Value
CRP	1.107 (1.033–1.186)	<0.01	1.050 (0.974–1.132)	0.20	−	−
AVG	1.143 (1.056–1.236)	<0.01	1.139 (1.052–1.233)	<0.01	1.050 (0.964–1.144)	0.260
STS	1.158 (1.046–1.281)	<0.01	1.178 (1.056–1.313)	<0.01	1.127 (1.011–1.256)	0.03
SII	1.002 (1.002–1.003)	<0.01	−	−	1.002 (1.001–1.003)	<0.01

Abbreviations: MACEs, major adverse cardiac events; CRP, C-reactive protein; AVG, aortic valve gradient; STS, Society of Thoracic Surgeons Score; SII, Systemic immune-inflammation index.

## Data Availability

The data that support the findings of this study are available on reasonable request from the corresponding author.

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
