# Peer review of "Prognostic Value of Systemic Immune-Inflammation Index for Major Adverse Cardiac Events and Mortality in Severe Aortic Stenosis Patients after TAVI"

_medicina, 2021, doi:10.3390/medicina57060588_

Round 1
Reviewer 1 Report
Dear Editor,
I evaluate the revised version of this paper and I think that the authors well addressed previous comments. The paper improved very much.
Reviewer 2 Report
This is well revised according to my comments.
There are no further comments.
Reviewer 3 Report
The authors did he suggested improvements to he manuscript.
This manuscript is a resubmission of an earlier submission. The following is a list of the peer review reports and author responses from that submission.
Round 1
Reviewer 1 Report
To:
Editorial Board
Medicina
Title: “Prognostic value of systemic immune-inflammation index for major adverse cardiac events and mortality in severe aortic stenosis patients after TAVI”
Dear Editor,
I read this manuscript and I think that:
- The retrospective nature of the study is a limitation. Please discuss this point into a dedicated limitation section.
- MACE at 6 month follow-up is a short follow-up period. This should be discussed in the discussion section.
- Authors should provide information about the clinical background of the patients. Table 1 should be updated with the main co-morbidities of the patients as they can impact on results.
- Same considerations are for the pharmacological background of the patients. Please discuss and provide data.
- The role of care manager should also be considered as this professional figure may play an important role in this context. Please consider and discuss the paper from Ciccone MM et al. Vasc Health Risk Manag. 2010 May 6;6:297-305.
- The regression analysis should be updated in order to evaluate the role of confounding factors on final results.
Author Response
Dear Editor,
Thank you for the consideration of our paper for publication in the Journal of Medicina. We read the reviewer’s valuable comments and constructive criticisms with great interest and we would like to resubmit our revised paper as instructed for possible publication in your esteem journal. Following revisions have been made in the paper according to editor and reviewer’s comments.
Reviewer 1
Dear Editor,
I read this manuscript and I think that:
Q1- The retrospective nature of the study is a limitation. Please discuss this point into a dedicated limitation section.
A1- This major limitation was mentioned in a dedicated limitation section.
Q2- MACE at 6-month follow-up is a short follow-up period. This should be discussed in the discussion section.
A2- This major limitation was mentioned in a dedicated limitation section.
Q3- Authors should provide information about the clinical background of the patients. Table 1 should be updated with the main co-morbidities of the patients as they can impact on results.
A3- We updated our Table 1 with the main co-morbidities of the patients.
Q4- Same considerations are for the pharmacological background of the patients. Please discuss and provide data.
A4- The pharmacological background of the patients was included in Table 1. According to our analysis, medical therapy had no effect on the results of the study.
Q5- The role of care manager should also be considered as this professional figure may play an important role in this context. Please consider and discuss the paper from Ciccone MM et al. Vasc Health Risk Manag. 2010 May 6;6:297-305.
A5- This information was mentioned in the text and this article was cited.
Q6- The regression analysis should be updated in order to evaluate the role of confounding factors on final results.
A6- The regression analysis was updated after including all possible confounders, which might effect of the results of the study.
Reviewer 2 Report
In the present manuscript, Dr. Aydin Rodi Tosu, et al demonstrated the prognostic impact of the systemic immune-inflammation index (SII) on TAVI. The authors demonstrated that SII was the independent predictor for worse outcomes following TAVI.
This result would be interesting for readers. However, there are some issues to consider:
Major issues
- SII is one of the formulas of inflammation. CRP is a conventional marker of inflammation. Authors should describe the better point of SII than CRP (e.g. AUC compare).
- Is there a difference between balloon-expandable valve and self-expandable valve? Inflammation patterns can be different between both. Furthermore, authors should mention the difference whether post-dilatation or not.
- In table3, authors used CRP and SII as a variable for multivariate analysis. Both were markers of inflammation. Authors should consider collinearity to analyze it. Authors should analyze separately.
Minor issues
- Page1, Lines 22
In the abstract, authors should revise “HR: 1002” to “HR: 1.002”.
- Page1, Lines 35
In the introduction part, authors should revise “80-89 years.1” to “80-89 years. [1]”.
- Page 4, Lines 140
In the table3, the authors should revise “aortic. valve gradient” to “aortic valve gradient” on the footnote.
- Page 6, Lines 158
In figure3, the authors should revise “High SII; 0 or 1” to “SII; high SII or low SII”.
Author Response
Dear Editor,
Thank you for the consideration of our paper for publication in the Journal of Medicina. We read the reviewer’s valuable comments and constructive criticisms with great interest and we would like to resubmit our revised paper as instructed for possible publication in your esteem journal. Following revisions have been made in the paper according to editor and reviewer’s comments.
Reviewer 2
In the present manuscript, Dr. Aydin Rodi Tosu, et al demonstrated the prognostic impact of the systemic immune-inflammation index (SII) on TAVI. The authors demonstrated that SII was the independent predictor for worse outcomes following TAVI. This result would be interesting for readers. However, there are some issues to consider:
Major issues
Q1- SII is one of the formulas of inflammation. CRP is a conventional marker of inflammation. Authors should describe the better point of SII than CRP (e.g. AUC compare).
A1- We compared the AUC values of CRP and SII. We noted that the AUC of SII in predicting MACEs was significantly higher than the AUC of CRP (AUC: 0.960 vs AUC: 0.714, respectively) (Figure 1).
Q2- Is there a difference between balloon-expandable valve and self-expandable valve? Inflammation patterns can be different between both. Furthermore, authors should mention the difference whether post-dilatation or not.
A2- There was no difference in both groups in terms of balloon-expandable valve, self-expandable valve, and post-dilatation or not. This information was added to Table 3.
Q3- In table3, authors used CRP and SII as a variable for multivariate analysis. Both were markers of inflammation. Authors should consider collinearity to analyze it. Authors should analyze separately.
A3- As suggested, we generated two multivariable Cox regression analysis models in order to prevent a multi-collinearity between SII and CRP. Interestingly, CRP was not independently related with MACEs according to our multivariate regression analysis.
Minor issues
Q1- Page1, Lines 22, In the abstract, authors should revise “HR: 1002” to “HR: 1.002”.
A1- It was corrected.
Q2- Page1, Lines 35, In the introduction part, authors should revise “80-89 years.1” to “80-89 years. [1]”.
A2- It was corrected.
Q3- Page 4, Lines 140, In the table3, the authors should revise “aortic. valve gradient” to “aortic valve gradient” on the footnote.
A3- It was corrected.
Q4- Page 6, Lines 158
In figure3, the authors should revise “High SII; 0 or 1” to “SII; high SII or low SII”.
A4- It was corrected as requested.
Reviewer 3 Report
This study aimed to evaluate whether the immune inflammation index might have a prognostic value for MACEs, including stroke, re-hospitalization, and short-term all-cause mortality at 6-months, in patients with aortic stenosis undergoing TAVR.
line 115 "and SII [1283 (IQ= 1153-1399) vs 522 (IQ = 344-845), p < 0.01]" it is not clear what the authors wanted to write here.
Table 1. Demographic and clinical parameters of the study cohort. In my opinion, this table should be divided into 2 separate tables. One with demographic characteristics and another one with clinical parameters.
Table 2. Procedural and post-procedural complications of all cases according to the SII. In my opinion, this table should be divided into 2 separate tables, one with procedural complications and another one with post-procedural complications.
" [n = 17 cases (44.7%) vs n = 1 cases (1.2%), n = 8 cases 127 (21.1%) vs n = 1 cases (1.2%), n = 6 cases (15.8%) vs n = 1 cases (1.2%), n = 9 cases (23.7%) 128 vs n = 0 cases (0%), and n = 7 cases (18.4%) vs n = 1 cases (1.2%), p < 0.01, respectively]. " This part of the results should be reorganised as it ist not clear what the authors wanted to express here.
Units of measurement should be mentioned in Figure 3.
Authors should discuss more their own results in the discussion section.
Author Response
Dear Editor,
Thank you for the consideration of our paper for publication in the Journal of Medicina. We read the reviewer’s valuable comments and constructive criticisms with great interest and we would like to resubmit our revised paper as instructed for possible publication in your esteem journal. Following revisions have been made in the paper according to editor and reviewer’s comments.
Reviewer 3
This study aimed to evaluate whether the immune inflammation index might have a prognostic value for MACEs, including stroke, re-hospitalization, and short-term all-cause mortality at 6-months, in patients with aortic stenosis undergoing TAVR.
Q1- line 115 "and SII [1283 (IQ= 1153-1399) vs 522 (IQ = 344-845), p < 0.01]" it is not clear what the authors wanted to write here.
A1- It was corrected as ‘Remarkably, patients with MACEs had also higher SII than those without MACEs [1283 (IQ= 1153-1399) vs 522 (IQ = 344-845), p < 0.01, respectively].’
Q2- Table 1. Demographic and clinical parameters of the study cohort. In my opinion, this table should be divided into 2 separate tables. One with demographic characteristics and another one with clinical parameters.
A2- It was divided into two parts as requested.
Q3- Table 2. Procedural and post-procedural complications of all cases according to the SII. In my opinion, this table should be divided into 2 separate tables, one with procedural complications and another one with post-procedural complications.
A3- It was divided into two parts as requested.
Q4- [n = 17 cases (44.7%) vs n = 1 cases (1.2%), n = 8 cases 127 (21.1%) vs n = 1 cases (1.2%), n = 6 cases (15.8%) vs n = 1 cases (1.2%), n = 9 cases (23.7%) 128 vs n = 0 cases (0%), and n = 7 cases (18.4%) vs n = 1 cases (1.2%), p < 0.01, respectively]. " This part of the results should be reorganized as it is not clear what the authors wanted to express here.
A4- It was corrected just as’ Patients with high SII had higher incidence of major vascular complications, major bleeding, permanent pacemaker implantation, re-hospitalization, post-procedural ischemic stroke or transient ischemic attack than those with low SII.’ in order to not causing confusing for the readers.
Q5- Units of measurement should be mentioned in Figure 3.
A5- It was added.
Q6- Authors should discuss more their own results in the discussion section.
A6- More information regarding to the results of the study were added into the discussion section.